# Trusted Aggregation (TAG): Model Filtering Backdoor Defense In Federated Learning

## Abstract

Federated learning is a framework for training machine learning models from multiple local data sets without access to the data in aggregate. A shared model is jointly learned through an interactive process between server and clients that combines locally learned model gradients or weights. However, the lack of data transparency naturally raises concerns about model security. Recently, several state-of-the-art backdoor attacks have been proposed, which achieve high attack success rates while simultaneously being difficult to detect, leading to compromised federated learning models. In this paper, motivated by differences in the outputs of models trained with and without the presence of backdoor attacks, we propose a defense method that can prevent backdoor attacks from influencing the model while maintaining the accuracy of the original classification task. TAG leverages a small validation data set to estimate the largest change that a benign user's local training can make to the output layer of the shared model, which can be used as a cutoff for returning user models. Experimental results on multiple data sets show that TAG defends against backdoor attacks even when 40% of the user submissions to update the shared model are malicious.

## 1 Introduction

Federated learning (FL) is a potential solution to constructing a machine learning model from several local data sources that cannot be exchanged or aggregated. As mentioned in Mahlool & Abed (2022), these restrictions are essential in areas where data privacy or security is critical, including but not limited to healthcare. Also, FL is valuable for companies that shift computing workloads to local devices. Furthermore, these local data sets are not required to be independent and identically distributed. Hence, a shared robust global model is desirable and, in many cases, cannot be produced without some form of collaborative learning. Under the FL setting, local entities (clients) submit their locally learned model gradients and weights to be intelligently combined by some centralized entity (server) to create a shared and robust machine learning model.

Concerns have arisen that the lack of control or knowledge regarding the local training procedure could allow a user, with malicious intent, to create an update that compromises the global model for all participating clients. An example of such harm is a backdoor attack, where the malicious users try to get the global model to associate a given manipulation of the input data, known as a trigger, with a particular outcome. Some methods (Kurita et al., 2020; Qi et al., 2020; Li et al., 2021) have been proposed to detect the triggers in the training data to defend against backdoor attacks. However, in FL, as only the resulting model gradients or weights are communicated back, such methods cannot be applied to defend against backdoor attacks. Furthermore, since the model update in FL assumes no access to all clients' data, there is less information available to help detect and prevent such malicious intent. Thus backdoor attacks may be easier to perform and harder to detect in FL.

In this paper, we first observe that the output distributions of models created by malicious users are very different from that of benign users. Specifically, there exists a discernible difference between malicious and benign user distributions for the target label class. Therefore, we can leverage this difference to detect backdoor attacks. Figure 1 shows a clear difference between models trained with and without a backdoor attack in the output scores for the target class on clean data. Therefore using a small clean data set, the centralized server can produce a backdoor-free locally trained model, which we will refer to as the trusted user. Another candidate user model can be compared on this small clean data set and excluded if they have an unusual outputs.

Motivated by the finding that the output distributions of a model with and without a backdoor are different, we propose comparing user and trusted models by the distributional difference between their outputs and the most recent global model to identify malicious updates. We use the trusted user to estimate the most considerable distributional difference a benign user's local training could produce and eliminate returning models that exceed this distance cutoff. This proposed method is effective against multiple state-of-the-art backdoor attacks at different strength levels. Even in the unreasonable setting where 40% of the clients are malicious for each update, our proposed method can achieve a similar model performance and eliminate backdoor attacks, which dramatically outperforms current alternative methods. In the experiment section, we demonstrate our method's ability on several data sets to prevent backdoor attacks. Additionally, the method performs well even when the attack happens every round and starts at the beginning of the federated learning process. Furthermore, our method does not affect the performance of the global model on clean data, resulting in no decrease in the accuracy of the original classification task.

## 2 RELATED WORK

**Federated Learning.** Federated learning (FL) is an emerging machine learning paradigm that has seen great success in many fields (Ryffel et al., 2018; Hard et al., 2018; Bonawitz et al., 2019). At a high level, FL is an iterative procedure involving rounds of model improvement until it meets some criteria. These rounds send the global model to users and select a subset of users to update the global model. Then those chosen users train their local copy of the model, and their resulting models are communicated back and aggregated to create a new global model. Typically, the final local model's gradients or weights are transmitted back to ensure data privacy.

**Backdoor Attack.** Recently, several backdoor attacks have been proposed to take advantage of the FL setting. In Xie et al. (2020), the authors show that the multiple-user nature of FL can be exploitable to make more potent and lasting backdoor attacks. By distributing the backdoor trigger across a few malicious users, they could make the global model exhibit the desired behavior at higher rates and for many iterations after the attack had concluded. We will show our threshold's effectiveness holds even when the backdoor attack is more frequently present in federated learning rounds than in the original paper. A recent work (Zhang et al., 2022) proposed a projection method, Neurotoxin, which claimed to increase the duration that a backdoor association remains present in the shared model after an attack has occurred. The attacker's updates are projected onto dimensions with small absolute values of the weight vector. The authors claim such weights are updated less frequently by other benign users, resulting in greater longevity of successful attacks. We will demonstrate our method's effectiveness against both of the above attacks (Xie et al., 2020; Zhang et al., 2022).

**Defense.** The most popular aggregation method of FL is FedAvg (McMahan et al., 2016). However, Median and Trim-mean, two other robust defense methods for FL, were proposed in Yin et al. (2018). The paper theoretically explores two robust aggregation methods: Median and Trim-mean, which were shown effective in defending against certain attacks in FL. Median is a coordinate-wise aggregation rule in which the aggregated weight vector is generated by computing the coordinate-wise median among the weight vectors of selected users. Trim-mean aggregates the weight vectors by computing the coordinate-wise mean using trimmed values, meaning that each dimension's top and bottom $k$ elements will not be used. We propose a method that can be implemented in addition to other aggregation or model filtering methods. Such other defense methods can be applied to the subset of the randomly selected users to update the model that our method returns. In the experiment, we focus on the original FedAvg (McMahan et al., 2016) aggregation to show the effectiveness of our proposed method without assistance from additional defense techniques.

Few defense methods have been proposed to defend against backdoor attacks in FL. Prior work (Shejwalkar et al., 2022) claims that norm clipping (Sun et al., 2019) is effective against backdoor attacks in FL but has been broken by the Neurotoxin attack. Recently proposed works for federated learning include (Rieger et al., 2022; Andreina et al., 2020). However, for our experiments, we focus on comparison with FLTrust (Cao et al., 2020) because it also relies on a small clean training data set. Although their original paper shows that their defense is effective against adaptive attacks where around half of the clients are malicious, we will demonstrate that their method fails against even the most straightforward backdoor attack that we consider in our experiments. Hence, our work is necessary for federated learning and an improvement over the current similar defense methodology.

## 3   TRUSTED AGGREGATION

This section describes the motivation and framework for our proposed method, Trusted Aggregation (TAG), which effectively defends against state-of-the-art backdoor attacks in federated learning. We will use the term output to refer to the last model layer's output, typically after several fully connected layers and before any softmax operation has occurred for classification models.

**Motivation.** We find that the output distributions of models returned by malicious users are very different from that of benign users. For an m-way classification problem, the outputs of the deep learning model are of the same dimensionality as the classification problem. The softmax operation is commonly performed on these outputs to produce predicted probabilities for each class. Hence, each element of the model output has a correspondence with exactly one class. This node-class association is specific to the outputs and may not exist for outputs of other hidden layers. For targeted backdoor attacks, they intend to produce an additional learned association between a given manipulation of the input data (trigger) and a specific class label (target). Hence, we believe intuitively that targeted backdoor attacks can be viewed as an (m + 1)-way classification problem. We believe such a difference in the model tasks may be exploited to identify models trained with a backdoor attack. In Figure 1, we observe that this learned association can come with a distributional change in the output scores specifically for the target class. We conclude that models with a backdoor may produce different distributions of output scores for clean data. Therefore it implies that if we had one model believed to be trained without a backdoor attack, we might identify whether another candidate model has a backdoor attack by comparing their outputted values on the same clean data.

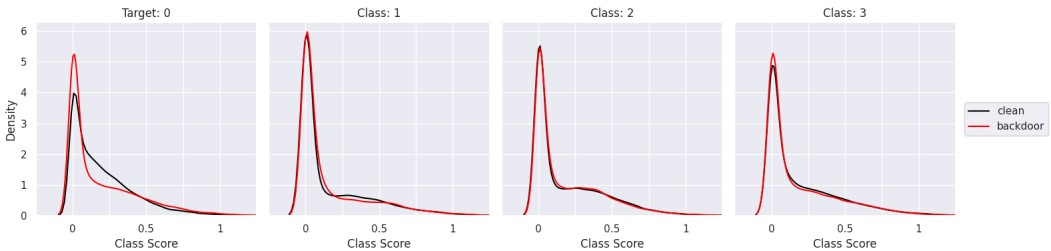

Figure 1: Output distributions (kernel density estimation based) conditional on class label for a **backdoor** model (red) and a **clean** model (black). Note the obvious difference between the distributions of the backdoor and clean models for the target label class.

**Detection Framework.** We assume that there exists a small clean validation data set that can be used for gatekeeping the global model for updates. Note that this data set can either belong to a trustworthy existing user or be collected by the centralized server and treated as a new user. Hence, we will refer to this trusted validation data as the trusted user. The detection method leverages the trusted user to evaluate incoming model weights and determine whether each contribution can participate in the global model update procedure. The main idea is to detect user models with an unusually distributed outputs using the clean data set from the trusted user. Our method can be easily extended when several trustworthy data sets or users are present.

See Figure 2 for an overview of our proposed detection framework. In each communication round, this validated user completes the following steps to generate a threshold for malicious user detection. First, the validation data is used to update a copy of the global model simultaneous to the local training of other users. When models are returned by the subset of users chosen to potentially participate in the update of the shared model, all models, including the validation user model and the global model, have predicted scores made and stored for the validation data. We denote the saved predicted scores as $\boldsymbol{o}_G, \boldsymbol{o}_T$, and $\boldsymbol{o}_j$ for the global, validation, and jth user models, respectively.

Next, we compute the distance between the trusted and other user models from the global model. Specifically, for each class c, we compute the class-conditional distributional distance $\mathcal{D}(v_j^{(c)}, v_T^{(c)})$ between the global model output ($\boldsymbol{o}_G^{(c)}$) and the user output ($\boldsymbol{o}_j^{(c)}$ or $\boldsymbol{o}_T^{(c)}$) by applying a distributional difference function. In our experiments, we use the Kolmogorov-Smirnov (KS) distance from estimated CDFs based on $\boldsymbol{o}_G^{(c)}, \boldsymbol{o}_j^{(c)}$, and $\boldsymbol{o}_T^{(c)}$. However, we believe there may be other applications where other distance functions should be successful. Suppose there are $m$ classes in total; the process will result in a distance vector ($\boldsymbol{v}_j, \boldsymbol{v}_T \in \mathbb{R}^m$) for each user, including the trusted user. The distance

vectors will then determine which users can participate in the update. Further details are given in Algorithm 1.

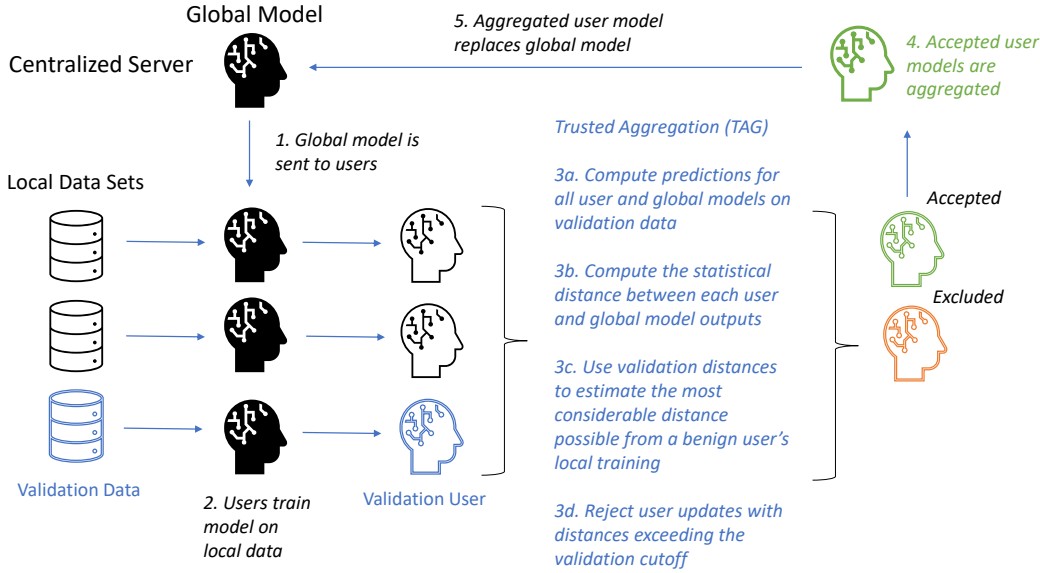

Figure 2: Diagram representation of our trusted aggregation detection framework.

---

**Algorithm 1** Trusted Aggregation

Notation: Let $\boldsymbol{S}$ represent the random subset of users that will submit locally trained models $U_j$ to update the global model $G$, $U_T$ to denote the model from the trusted user, $\boldsymbol{X}$ to denote the local data of the trusted user, $\mathcal{D}$ to represent the distributional difference function, and $\theta \geq 1$ for the method's scaling coefficient.

---

1: **procedure** TRUSTED AGGREGATION($\boldsymbol{X}, G, U_T, \{U_j\}_{j \in \boldsymbol{S}}, \theta$)
2:      Generated outputs: $\boldsymbol{o}_G = G(\boldsymbol{X})$, $\boldsymbol{o}_T = U_T(\boldsymbol{X})$, and $\boldsymbol{o}_j = U_j(\boldsymbol{X})$, $\forall j \in \boldsymbol{S}$
3:      **for** each class $c \in [1, ..., m]$ **do**
4:          Compute the distributional distances between each user and the global model
5:          $v_T^{(c)} = \mathcal{D}(\boldsymbol{o}_G^{(c)}, \boldsymbol{o}_T^{(c)})$ and $v_j^{(c)} = \mathcal{D}(\boldsymbol{o}_G^{(c)}, \boldsymbol{o}_j^{(c)})$, $\forall j \in \boldsymbol{S}$     $\triangleright \, \boldsymbol{o}^{(c)}$: output for class $c$
6:      **end for**
7:      The above procedure produces: $\boldsymbol{v}_T \in \mathbb{R}^m$, $\boldsymbol{v}_j \in \mathbb{R}^m$     $\triangleright \, m$: total number of classes
8:      Compute threshold: $\tau_r = \theta \times \max(\boldsymbol{v}_T)$     $\triangleright \max$: maximum element of the vector
9:      $\tilde{\tau} \leftarrow$ GLOBAL-MIN MEAN SMOOTHING($\tau$)     $\triangleright$ Algorithm 2
10:     Select users: $\boldsymbol{S}_r = \{j \in \boldsymbol{S} \,|\, \max(\boldsymbol{v}_j) < \tilde{\tau}\}$
11:     **return** FedAvg($\{U_j\}_{j \in \boldsymbol{S}_r}$)
12: **end procedure**

---

**Threshold Construction.** In this part, we discuss how to decide the threshold ($\tau$) and how to use it to select users. We define the true threshold as the largest possible change a non-malicious user could contribute. Users with distance values exceeding the threshold should be excluded. Naively, this can be thought of as an estimation of the maximum given a single sample. Hence, the true maximum will be at least as large as our observed value. Therefore, we parameterize our method by $\theta \geq 1$, see Algorithm 1.

In the case that the class-conditional distances ($v^{(c)}$) are Uniform on $[0, b_c]$ for each class $c$, where $b_c$ is the maximum possible change to the output layer of class $c$ through local training by a non-malicious user. Hence, estimating the threshold consists of estimating the maximum of $b_c$ for any class. Let $m$ represent the total number of classes. Proposition 1 gives simple bounds for our quantity of interest under arbitrary dependence.

**Proposition 1** *If $v^{(c)} \sim Uniform(0, b_c), \forall c \in [1, \ldots, m]$ then $E\left[V\right] \leq b_j \leq E\left[2V\right]$ where $V = \max_c v^{(c)}$ and $j = \arg\max_c (b_c)$. Proof in Appendix C.1.*

However, we acknowledge that it may be unreasonable to assume that class conditional distances are Uniform as many training hyper-parameters and even model choice will impact the distance distributions. We present these results to assist in understanding reasonable magnitudes for $\theta$ as many distributions may not require large scaling values ($\theta \in [1, 2]$ for Uniform). In general, we recommend choosing $\theta$ based on the setting's prevalence of backdoor attacks (potentially unknown) and the cost of a successful attack to interested parties. Relatively large values for $\theta$ will allow more users to update the global model but increase the risk of a successful backdoor attack. Conversely, using our threshold $\tau$ without scaling would help to prevent stronger backdoor attacks, but with the potential loss of denying benign users from updating the global model. The goal is to choose the smallest $\theta$ that allows benign users to sufficiently create a robust shared machine learning model.

Since the validated user is non-malicious, their distance vector serves as a good representation for other non-malicious users. Therefore, we estimate the threshold $\tau$ with $\theta \times \max(\boldsymbol{v}_T)$, where $\boldsymbol{v}_T \in \mathbb{R}^m$ is the distance vector of the validation user and $\max(\cdot)$ means getting the maximum value of the vector $\boldsymbol{v}_T$. Then, the maximum distance value ($\max(\boldsymbol{v}_j)$) of each selected user will be compared with the threshold ($\tau$) to determine the final list of users who can participate in the update. A user with a maximum distance smaller than the threshold is considered a benign user, while a user with a maximum distance larger than or equal to the threshold will be removed. However, this naive threshold is very unstable as a lucky malicious user can get past it in some rounds due to the instability. Therefore, we make an additional modification, **global-min mean smoothing**, to this basic threshold to address the concern.

**Global-Min Mean Smoothing.** A naive way to stabilize the threshold value is using a smoothing method, like a moving average. However, in the early communication rounds, the naive threshold value rapidly decreases as the model starts making connections between inputs and output classes. Therefore, applying a traditional smoothing method can result in a relatively high threshold early, which may let attackers bypass it. When the naive threshold ($\tau$) decreases rapidly, we do not wish to use any previous communication rounds for the smoothing.

Therefore, we propose to use the lowest observed value (Global Min) of $\tau$ as the starting point of the smoothing window. Let $\tau_t$ represent the naive estimation of the threshold in round $t$. Then the smoothed threshold $\tilde{\tau}$ at round $n$ is given by

$$\tilde{\tau} = \frac{1}{n - t_s + 1} \sum_{t=t_s}^{n} \tau_t,$$

where $t_s$ is the round that when the global min is observed. Details of the global-min mean smoothing are described in Algorithm 2. As $\tau_t$ shrinks, we observe new global minimums, and the start of the threshold smoothing is reset. In addition, when our estimates do not continue to fall, previous values are leveraged to smooth the cutoff, which keeps lucky malicious users from getting past a volatile threshold. Figure 3 compares our global min-

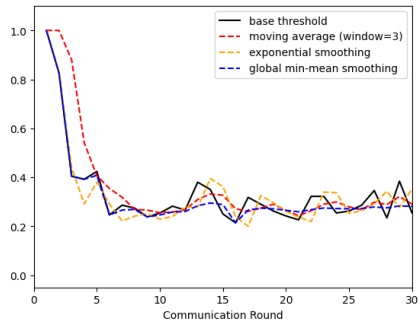

Figure 3: Comparison of the global min-mean smoothing with the base (naive) threshold and various smoothing methods.

mean smoothing with the naive threshold and various smoothing techniques. The global min-mean smoothing best captures the naive threshold's early behavior while providing remarkable stability improvements. Most importantly, when our threshold encounters a new global minimum, it provides a conservative estimate to prevent malicious users while re-learning cutoff behavior over the next few rounds.

---

**Algorithm 2** Global-Min Mean Smoothing

Notation: Let $(\tau_1, \cdots, \tau_{n-1}, \tau_n)$ denote the sequence of values that we wish to smooth.

---

1: **procedure** GLOBAL-MIN MEAN SMOOTHING($\tau_1, \cdots, \tau_{n-1}, \tau_n$)
2:     Record the location of global minimum: $i = \underset{t \in [1,...,n]}{\arg\min} \tau_t$
3:     Subset to a sequence starting with the global min: $\{\tau_t\}_{t=i}^n = \{\tau_i, \cdots, \tau_n\}$
4:     **return** average of sequence subset, $\overline{\{\tau_t\}_{t=i}^n}$
5: **end procedure**

---

# 4 EXPERIMENTS

In this portion, we provide the following experiments and conclusions. First, in Section 4.2 we show that TAG is the only considered effective defense against several strong backdoor attacks. Secondly, we demonstrate in Section 4.3, that smoothing techniques are in fact necessary for our unstable round-to-round threshold. Next, Section 4.4 extends our main results to when users have non-independent and non-identically distributed data sets, showing the effectiveness of our method for various applications of federated learning. Finally, we include Section 4.5 to emphasize that our method is robust to changes to the distribution and size of the trusted user's validation data. We conclude through extensive experimentation that Trusted Aggregation provides a useful advancement toward model security in the federated learning environment.

## 4.1 SETTING

**Federated Learning.** We start by giving further specifications regarding the federated learning environment. Our interest is training a global model over $M$ communication rounds with $N$ users. Each iteration randomly selects $K$ users, using a specified proportion of the total users, to participate in the model update. See Section A for further experimental specifications.

After local training, the next global model is the average returned model weights by the FedAvg procedure. In our experiments, we use ResNet18, a popular classifier proposed in He et al. (2016). Additionally, in Section B.3, we demonstrate that our success in not model dependent by repeating main results using VGG16 (Simonyan & Zisserman (2014)). We assume that all users, including malicious, have complete control over all aspects of local training, such as learning rate, the number of epochs, and the model weights they return. For simplicity, we select two main sets of training hyper-parameters for benign and malicious users. The malicious users will poison a given proportion of their local data by adding their backdoor trigger to the input and changing the training label to the target class. They intend for the model to associate the trigger with the target class and hence have the future global model identify any input with the trigger as belonging to the target class.

**Attack and Baseline.** To show the effectiveness of our method, we choose a setting in which the backdoor attack is powerful. We force the same malicious users to be included in the subset of selected users to update the global model each round after the start of the backdoor attack. Note that the selection of random users is a defense against malicious users by making it difficult for them to update the global model repeatedly. Additionally, we do not allow the validation user, a guaranteed benign user, to participate in any global model updates. We make these decisions to highlight the ability of our threshold to prevent even unreasonably strong backdoor attacks against the global model. For our experiment, we test the proposed method, Median and Trim-mean (Yin et al., 2018), and FLTrust (Cao et al., 2020) against two state-of-the-art backdoor attacks in FL: Neurotoxin (Zhang et al., 2022) and Distributed Backdoor Attacks (DBA) (Xie et al., 2020). To further evaluate the effectiveness of the aggregation methods, we also vary the proportion of malicious attackers to comprise between 10% and 40% of the selected users to test the defense methods under different attack strength levels.

**Data.** The experiments are done on three different data sets: CIFAR10 (Krizhevsky & Hinton, 2009), CIFAR100 (Krizhevsky & Hinton, 2009), and STL10 (Coates et al., 2011). In each experiment, we construct user data sets by random sampling from the training data. Again, see Appendix A for further experiment details and hyperparameters. For global model evaluation, we split the test set into two parts. We add the backdoor trigger to images in the second half and remove any target class observations. We measure model performance on the first half using classification accuracy, and the proportion of the poisoned half predicted as the target class, known as attack success rate, to measure the extent that the backdoor attack has compromised the model. For a defense method, a good performance consists of a low attack success rate and high classification accuracy. In other words, attacks are unsuccessful when the defense method is used, and the defense does not negatively influence the classification performance.

## 4.2 COMPARISON OF DEFENSE METHODS AGAINST BACKDOOR ATTACKS

We begin by considering a setting where 10% of the returning user models are malicious each communication round. Figure 4 shows the performance of the various methods against backdoor attacks with and without the Neurotoxin projection on three data sets regarding model classification

accuracy and attack success rate. For our main results, we use the TAG scaling coefficients ($\theta$) of 2, 2, and 1.1 for the data sets CIFAR10, CIFAR100, and STL10, respectively. Our proposed method nullifies the backdoor attack in each case without a meaningful decrease to the classification accuracy of the original task in general. However, the other methods, coordinate-wise Median, Trim-mean, and FLTrust, fail to prevent backdoor attacks with or without Neurotoxin. Although FLTrust was shown in its original paper to be successful against an adaptive (attacker knows defense) data poisoning attacks aiming to change the global model update along the opposite direction of the global model update under no attacks, FLTrust fails to prevent targeted backdoor attacks in the weakest attack setting we consider. We conclude that our method is a clear improvement to these existing defense methods for federated learning.

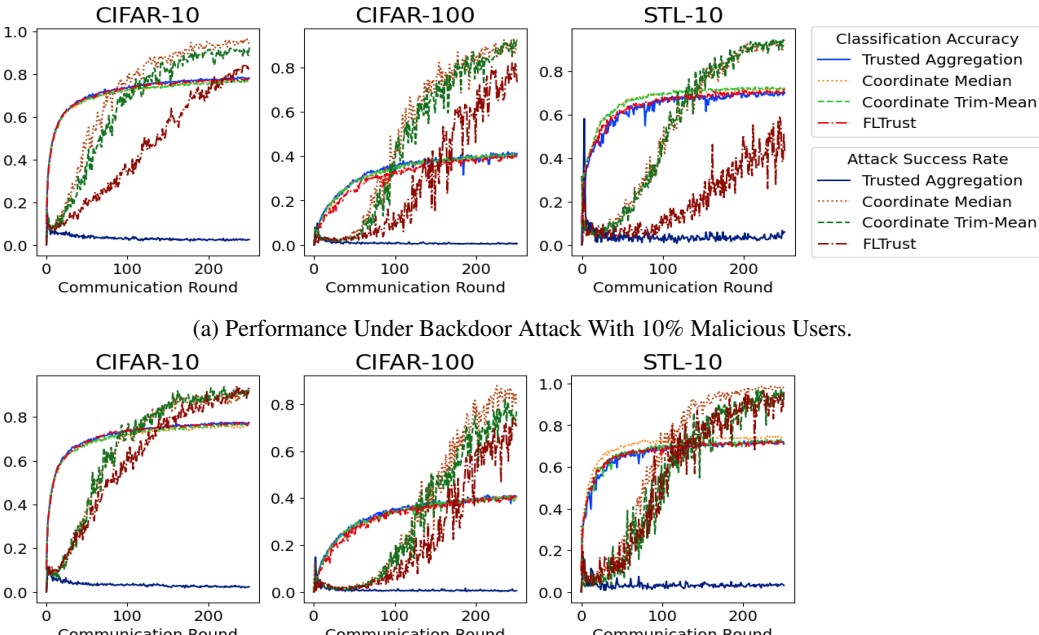

(a) Performance Under Backdoor Attack With 10% Malicious Users.

(b) Performance Under Neurotoxin Backdoor Attack With 10% Malicious Users.

Figure 4: Model performance under standard (*top*) and Neurotoxin (*bot*) backdoor attacks with 10% malicious users. The proposed method TAG performs well in defending against backdoor attacks as the attack success rates are low. Meanwhile, it does not generally affect the model's classification performance on clean data, except for a slight decrease on STL10 for the DBA attack. However, the other methods do not work well against any backdoor attacks.

TAG can handle more difficult attack settings against state-of-the-art attacks. We observe similar results compared to our other attack settings when testing the defense methods against DBA and Neurotoxin attacks where 20% of updates are malicious. Unsurprisingly, the other defenses, which failed against the previous setting, cannot prevent stronger attacks whereas TAG again prevents all backdoor attacks. See Section B.1 for results.

We instead focus on even stronger DBA and Neurotoxin attacks with 40% malicious users in the selected set. These attacks are catastrophically successful against the current robust aggregation methods, see Figure 5, having a nearly perfect attack success rate after round 50 on all our data sets. In some settings FLTrust, is able to delay the attack success compared to robust aggregation, but still fails to prevent any of the backdoor attacks of this or previous strengths. However, our method, TAG, overcomes the backdoor extent of the initial rounds to prevent the attack against all data sets.

We conclude that even when the federated learning setting updates consist of almost half malicious users, we can use TAG to attempt to eliminate backdoor attacks. We show various additional experimentation results in our Appendix to strengthen the evidence for our claims. First, in Section B.2, we offer that our model does not hinder performance for the original classification task even without the presence backdoor attack compared to FedAvg. Secondly, Section B.3 supports that our main results, reproduced with VGG16 instead of ResNet18, are not dependent on model choice. Hence our method leads to increased model security at a negligible cost to the shared model, even under a lack

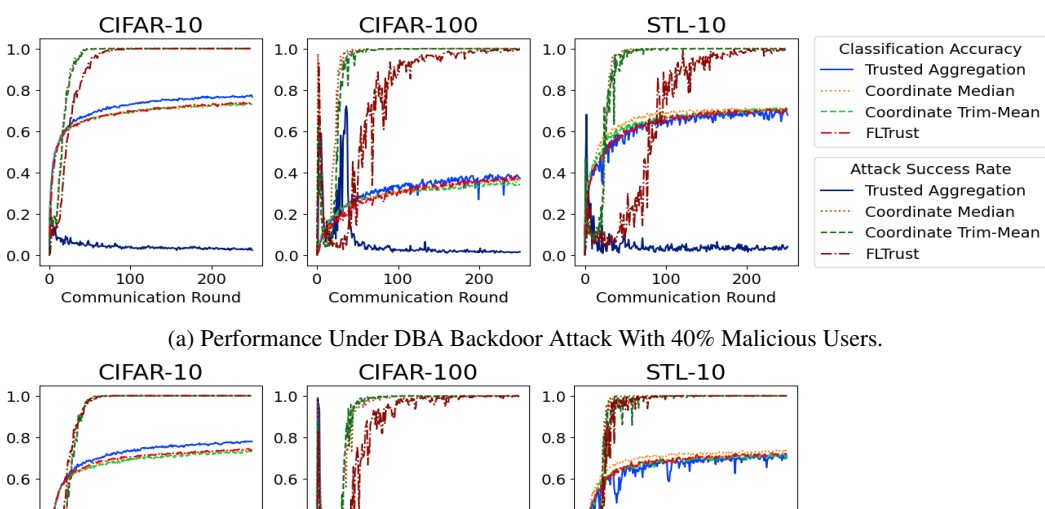

(a) Performance Under DBA Backdoor Attack With 40% Malicious Users.

(b) Performance Under DBA And Neurotoxin Backdoor Attacks With 40% Malicious Users.

Figure 5: Model performance under DBA without (*top*) and with (*bot*) Neurotoxin backdoor attacks with 40% malicious users. TAG is the only method that prevents the backdoor attacks.

of attack. We conclude that Trusted Aggregation (TAG) is an essential advancement toward model security for the federated learning framework.

### 4.3 NECESSITY OF THRESHOLD SMOOTHING

We revisit the last attack considered on the STL10 data set to highlight the importance of our proposed technique, global-min mean smoothing. Recall that our smoothing is intended to improve the stability of our estimated threshold while preserving its behavior in the initial rounds, not conserved by other smoothing techniques. If we repeat the backdoor attack where 40% of the user subset is malicious each iteration and omit the global-min mean smoothing, our method no longer can prevent the backdoor attacks with Neurotoxin projection, see Figure 6. Hence, without smoothing, we conclude that malicious users may have an opportunity to get past a less stable cutoff.

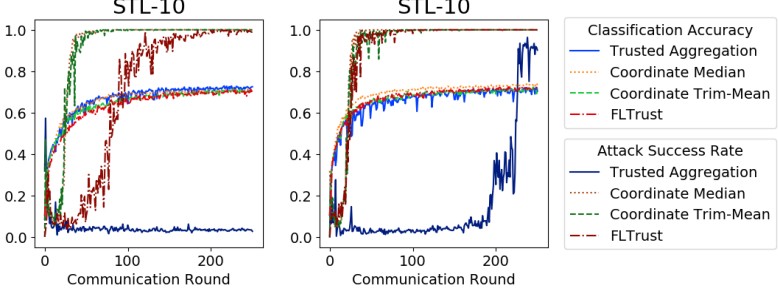

Figure 6: Model performance under DBA without (*left*) and with (*right*) Neurotoxin backdoor attacks with 40% malicious users on STL10 without using global-min mean smoothing. TAG fails to defend against the backdoor attack on STL10 without the improved stability of our estimate from our proposed smoothing technique.

### 4.4 EXTENDING RESULTS TO IMBALANCED USER DATA SETS

In this section, we show the applicability of our method for imbalanced users focusing on the CIFAR10 data set. Recall that local data sets are not required to be independent and identically distributed for federated learning. In this experiment, we use the m-dimensional (number of total classes) Dirichlet distribution with $\alpha = 1$ to determine the proportion for each class label to be

randomly sampled. Recall, we parameterize the Dirichlet distribution by the simplified $\alpha\mathbf{1}_m$ where $\mathbf{1}_m$ is an m-dimensional vector of ones, and $\alpha \in [0, \infty)$ is a scalar such that smaller values lead to imbalanced user data.

Regarding attacks with 10% and 20% malicious users, we did not have to tune $\theta$ (using $\theta$ from previously balanced data experiments) to have a very successful defense when data is highly imbalanced ($\alpha = 1$), see results in Section B.4 of the Appendix for further discussion. In summary, we repeat the previous experiment without modifying $\theta$ to understand how different defending against backdoor attacks, with TAG, is for imbalanced user data. TAG is reasonably robust to the weaker backdoor attacks without tuning $\theta$ for the new application. However, $\theta$ should be adjusted for each application to ensure TAG's best possible defense.

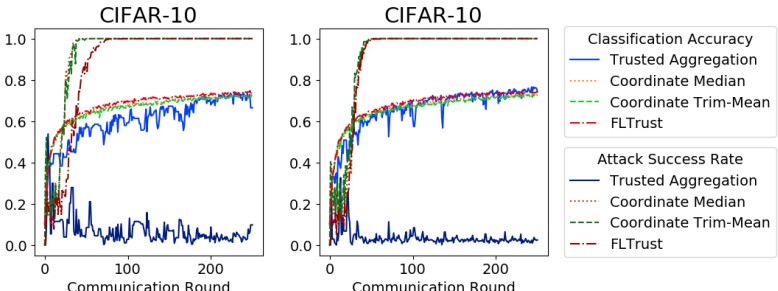

Figure 7: Model performance under DBA without (*left*) and with (*right*) Neurotoxin backdoor attacks with 40% malicious users on imbalanced local user data sets with tuned $\theta$. The proposed method, TAG, performs well in defending against backdoor attacks, even when the user data is imbalanced. Again, the other defense methods do not prevent any backdoor attack under imbalanced data.

When we do tune $\theta$, TAG is effective for all attack prevalences, including 40%. Recall the trade-off between robustness against backdoor attacks and performance for the original classification task. We attempt to choose the smallest $\theta$ that successfully trains a global model regarding classification accuracy. In Figure 7, with $\theta = 1.25$ and $\theta = 1.5$ respectively, TAG can eliminate the 40% malicious backdoor attacks. With less scaling, the original classification accuracy suffers in early communication rounds as fewer users update the model. However, as the global model nears convergence, the round-to-round accuracy stabilizes and becomes similar to or better than the baseline methods. Hence, we conclude that our proposed method, TAG, can handle backdoor attacks on imbalanced data and without losing accuracy for the original task. These experiments support our recommendation to choose the smallest scaling possible that achieves desired performance on the original task, and this setting is more than sufficient to show the usefulness of our method for the majority of federated learning applications.

## 4.5 ROBUSTNESS TO TRUSTED USER DATA SET

Recall that the trusted user either consists of a guaranteed non-malicious user or the centralized server collects a small validation data sets and acts as another user. We argue that the centralized server can take precautions to collect a representative clean data set. However, if the trusted user is an actual user or the centralized server fails to obtain such a representative sample, it is vital to understand how the data distribution of the trusted user impacts our defense. Additionally, we believe that we only need consider the case that the trusted data set is imbalanced if the user data sets are also imbalanced. When the users have access to balanced data, it seems plausible that our trusted user would also likely have access to balanced data. For this section, we consider the case where both our users and trusted user have data Dirichlet sampled data sets from CIFAR10 with $\alpha = 1$.

In Section B.5.1, we have results on model performance with imbalanced trusted user data without tuning $\theta$ from those obtained for the previous imbalanced user experiments. These additional experiments serve to show the difference in defending against backdoor attacks with and without a representative trusted user data set. In summary, we see similar results to robustness of TAG to the data distribution of users where we can defend against our weaker attack settings without tuning $\theta$. For TAG to be most effective, however, our scaling coefficient $\theta$ must be tuned for every application to provide its best defense. Additionally, Section B.5.2 shows that dramatically reducing the size of the trusted data set, relative to other users, does not impact the success of TAG as a defense method.

Furthermore, in all these experiments, FLTrust is allowed a full sized, representative trusted user data set, and yet is still greatly outperformed by TAG.

Tuning TAG's scaling coefficient for the 40% malicious update setting gives $\theta = 1.25$ and $\theta = 2$ for DBA without and with Neurotoxin, respectively. In Figure 8, we observe that even with an imbalanced trusted user data set and 40% of user updates are malicious, our proposed method, TAG, can effectively prevent backdoor attacks. We observe that in early communication rounds, classification accuracy for the original task suffers, but the model's performance at convergence is not impacted by the presence of our defense. Therefore, we conclude that although a representative trusted data set is desirable, TAG is an essential tool for backdoor defense even when a representative trusted data set is not obtained. TAG can be an effective defense regardless of whether any users, including the trusted user, have data that is non-independent and non-identically distributed.

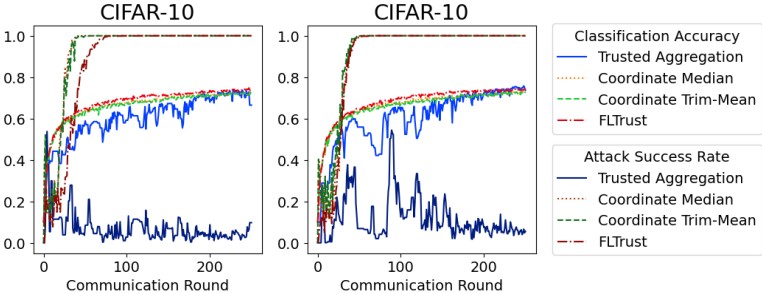

Figure 8: Model performance under DBA without (*left*) and with (*right*) Neurotoxin backdoor attacks with 40% malicious users on imbalanced local user data sets with an equally imbalanced trusted validation data set where $\theta$ is tuned.

## 5  LIMITATIONS

We demonstrate in Section 4.4 and Section 4.5 the success of TAG for non-independent and non-identically distributed imbalanced (non-iid) user data sets. First, we acknowledge that there exist many more extreme cases of non-iid data sets. For example, we do not test our method in the class partition case where each user holds the entirety of a small number of the total classes. However, we note that for more extreme data distributions, it may not be reasonable to consider learning a single shared model for all users, which is the main objective in this work. Second, although our framework is extendable to other models (regression, NLP, etc.), we only have experimental results for common classification computer vision models and databases. Future work may need to be done to modify our method or demonstrate success for other applications.

Most importantly, we do not claim that our defense can successfully defend against an adaptive attack. When the malicious attacker has knowledge that our defense is being used (adaptive attack), they can train two copies of the global model. They can train the first copy without data poisoning and use this model for penalizing the output layer distribution of the second copy while performing a backdoor attack. We reemphasize, in favor of our work, that as a solution our method can be used in conjunction with other defends methods and also is vastly outperforming a similar defense with claims towards adaptive attacks. Regardless, we believe future work is needed to modify our defense to work in the adaptive setting.

## 6  CONCLUSION

While current defense methods are failing to prevent even mild backdoor attack settings, TAG holds up against state-of-the-art attacks in unreasonably strong settings. Furthermore, TAG works against backdoor attacks when users have heterogeneous and imbalanced data and does not decrease the classification accuracy for the original task regardless of whether or not attacks are present. Furthermore, although our method relies on a some additional data, we show through experimentation that regardless of size or distribution TAG is robust to many attacks and effective against very strong attacks with proper tuning. Finally, TAG is compatible with other filtering methods or modifications to the choice of aggregation step and can be used in conjunction with other strong defense methods to prevent attacks. We conclude that TAG is an essential advancement toward model security for the federated learning framework.

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

## A  EXPERIMENTAL DETAILS

For reproducibility, we offer in Table 1 as a clear description of all arguments used in our experiments. In conjunction with our GitHub code at (LINK TBD), these details enable readers to repeat and expand upon our results.

It is important to note that for the STL10 data set, we inverted the data splits commonly referred to as train and test. We avoid using any unlabeled data, and we use the more extensive labeled data set (800 images per class) for training. Also not specified in the table, local user data sets are randomly sampled from the training data sets where a Dirichlet distribution determines the frequency of each class. Balanced local data sets are obtained using $\alpha = 10000$ to scale a vector of ones with dimensionality equal to the total number of classes. In certain experiments, imbalanced user and trusted data sets are created by the modification $\alpha = 1$.

|  | Hyperparameter | Variable Name | CIFAR10 | CIFAR100 | STL10 |
|---|---|---|---|---|---|
| Federated Learning | Users | $n\_users$ | 100 | 100 | 20 |
|  | Local Data Size | $n\_user\_data$ | 500 | 500 | 400 |
|  | User Subset Proportion | $p\_report$ | .1 | .1 | .5 |
| Data Augmentation | Padding |  | 4 | 4 | 12 |
|  | Random Horizontal Flip | NA | .5 | .5 | .5 |
|  | Random Crop Size |  | 32 | 32 | 96 |
| All Users | Batch Size | $n\_batch$ | 64 | 64 | 128 |
|  | Weight Decay | $wd$ | 5e-4 | 5e-4 | 5e-4 |
| Benign Users | Local Epochs | $n\_epochs$ | 10 | 10 | 10 |
|  | Learning Rate | $lr$ | .01 | .01 | .01 |
| Malicious Users | Local Epochs | $n\_epochs\_pois$ | 20 (15) | 15 | 25 (15) |
|  | Learning Rate | $lr\_pois$ | .01 | .01 | .005 (.01) |
| Data Poisoning | Poisoning Proportion | $p\_pois$ | .1 | .1 | .1 |
|  | Stamp Pixel Height | $row\_size$ | 4 | 4 | 24 |
|  | Stamp Pixel Width | $col\_size$ | 4 | 4 | 24 |
| Backdoor Defense | TAG Scaling ($\theta$) | $d\_scale$ | 2 | 2 | 1.1 |
|  | Trim Mean | $beta$ | .2 | .2 | .2 |

Table 1: Default arguments for all experiments unless otherwise specified. For all experiments, alternative values for $\beta$ did not seem to prevent the backdoor attacks. Any values modified for Neurotoxin attacks are shown in parentheses.

# B  Supplementary Results

## B.1  Comparison Of Defense Methods Against Backdoor Attacks (Continued)

TAG can handle difficult attack settings against state-of-the-art attacks. We consider testing the defense methods against DBA and Neurotoxin attacks with 20% malicious users in the selected set. Unsurprisingly, the other defenses, which failed against 10% malicious, cannot prevent stronger attacks, see Figure 9. We do observe that FLTrust appears to delay the attack success relative to robust aggregation but still results in a compromised model. On the other hand, our method, TAG, again prevents all backdoor attacks. For results demonstrating model performance against 10% and 40% malicious updates, see Section 4.2.

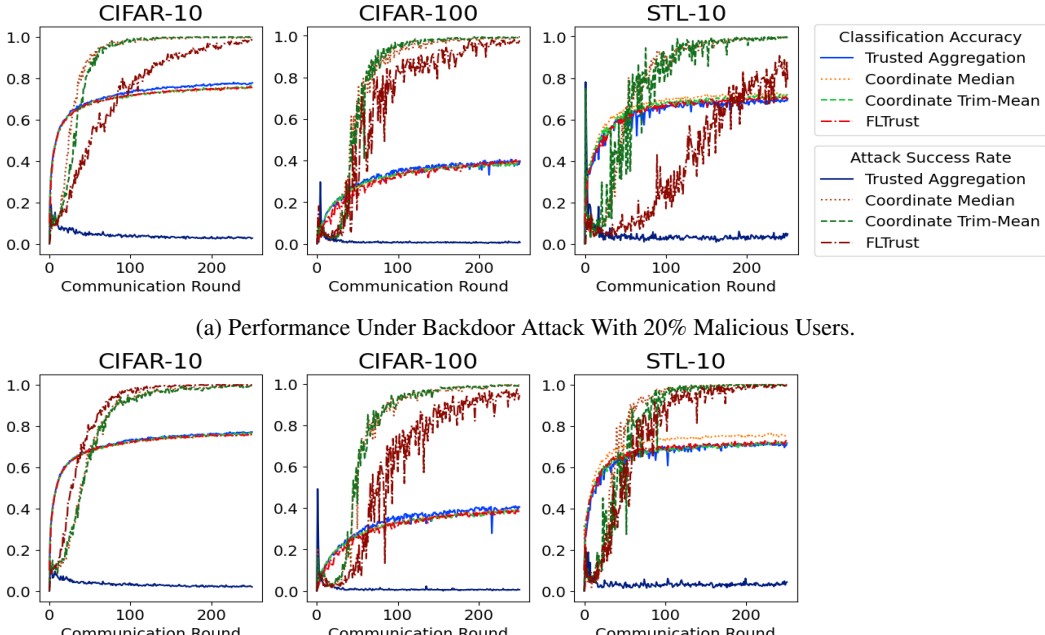

(a) Performance Under Backdoor Attack With 20% Malicious Users.

(b) Performance Under Neurotoxin Backdoor Attack With 20% Malicious Users.

Figure 9: Model performance under standard and Neurotoxin backdoor attacks with 20% malicious users. TAG is the only method that prevents these backdoor attacks.

## B.2    TAG Classification Accuracy Without Attackers

A successful backdoor defense can prevent attackers while best preserving the model performance on the original task. The model ability for the original task must also be maintained in the absence of attack. In addition to successfully preventing attacks when present, we observe in Figure 10 that our defense does not hinder the classification accuracy of the original task on STL10 compared to the FedAvg procedure. This supports the overall usefulness of our proposed method as the shared model will have improved security with our defense without cost if the system is not threatened by attacks.

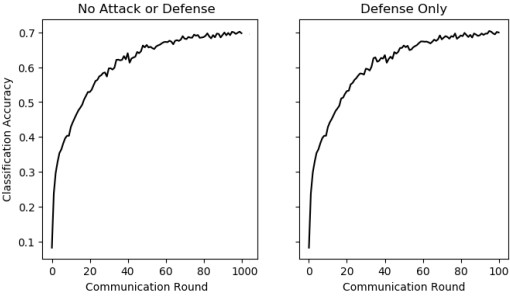

Figure 10: Model classification accuracy on STL10 in the absence of backdoor attacks without (*left*) and with (*right*) TAG defense.

### B.3 MODEL GENERALIZABILITY

In this section, we demonstrate that our results are not architecture dependent by repeating our main experiment results on CIFAR10 for another off-the-shelf image classification model. The following results are obtained using VGG16 with batch normalization, originally proposed in (Simonyan & Zisserman, 2014). In this experiment, Trim Mean is parameterized by $\beta = 0.1$. However, other values for $\beta$ did not impact defense.

Recall that we attempt to choose the smallest scaling coefficient $\theta$ such that the global model obtains desirable classification accuracy. For CIFAR10, $\theta = 2$ is successful for the ResNet18 architecture both for preventing backdoor attacks and for good classification accuracy for the original task. However, VGG required $\theta = 2.5$ to train a sufficient global model under Neurotoxin attacks. We observe that the choice of $\theta$ may depend on many hyperparameters of various parts of the federated learning procedure. In all attack settings, as presented in our main results, Section 4.2, our proposed defense TAG is the only method successful in preventing backdoor attacks. We again note that the successful defense of TAG is not associated with a meaningful change in global model performance on the original classification task. We conclude that TAG can be an effective backdoor defense for federated learning for various model choices.

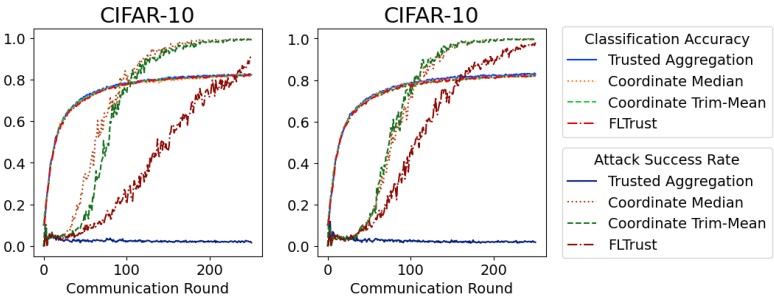

Figure 11: VGG model performance under standard (*left*) and Neurotoxin (*right*) backdoor attacks with 10% malicious users.

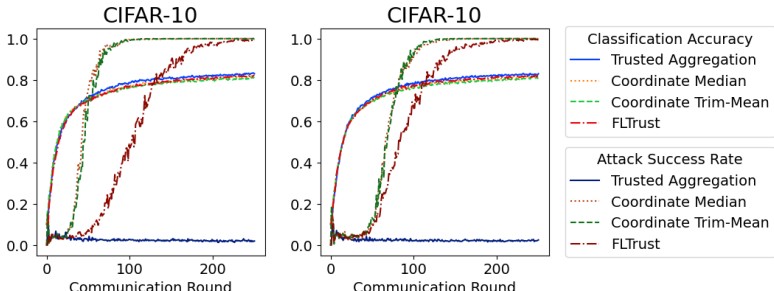

Figure 12: VGG model performance under DBA without (*left*) and with (*right*) Neurotoxin backdoor attacks with 20% malicious users.

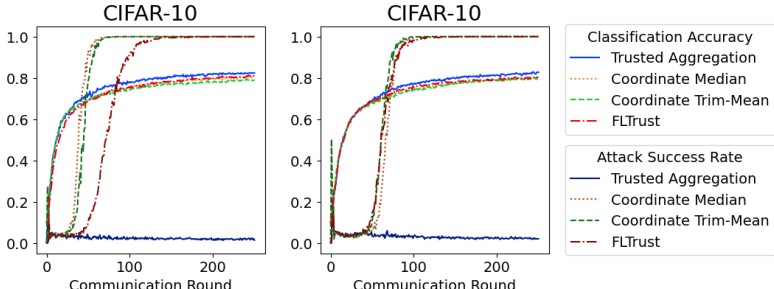

Figure 13: VGG model performance under DBA without (*left*) and with (*right*) Neurotoxin backdoor attacks with 40% malicious users.

## B.4 EXTENDING RESULTS TO IMBALANCED USER DATA SETS (CONTINUED)

We present the omitted figures from Section 4.4 to understand the success of backdoor attacks when local user data sets are imbalanced. Figure 14 and Figure 15 show that even without tuning TAG's scaling coefficient $\theta$, our proposed defense is effective for imbalanced user data. Note that we are using $\theta = 2$ as obtained from tuning our defense method on balanced local user data sets for CIFAR10. Similar to other experimentation results, TAG is the only method to prevent our backdoor attacks without changing model performance for the original classification task.

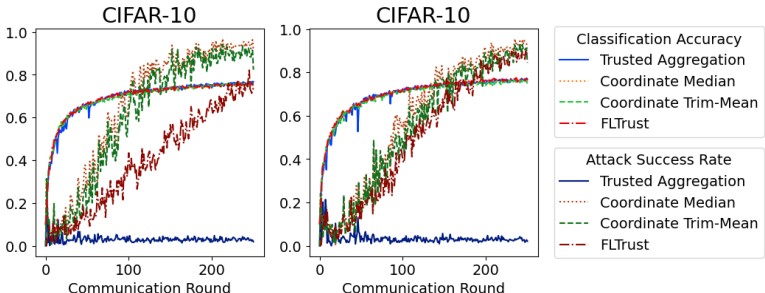

Figure 14: Model performance under standard (*left*) and Neurotoxin (*right*) backdoor attacks with 10% malicious users and imbalanced local user data sets without tuning $\theta$.

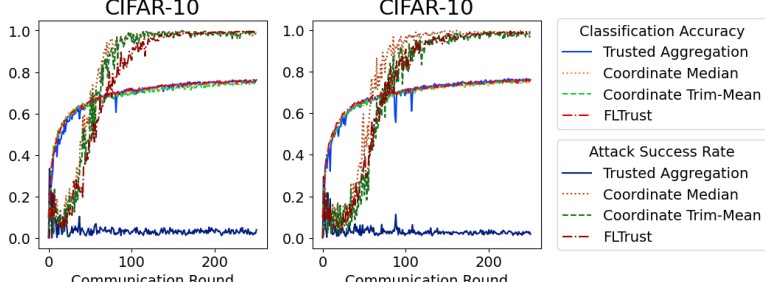

Figure 15: Model performance under DBA without (*left*) and with (*right*) Neurotoxin backdoor attacks with 20% malicious users and imbalanced local user data sets without tuning $\theta$.

This section's results help us understand how different backdoor defense is under other local user data distributions and the implications of misspecified TAG scaling. Consistently TAG scaling is robust to changes under weaker attacks but needs application-specific tuning to offer its best backdoor defense. If $\theta$ is not tuned, both considered backdoor attacks are successful at 40% prevalence, see Figure 16. However, in Section 4.4, we observe that with proper tuning of $\theta$, even with imbalanced user data, TAG can prevent the powerful attack where 40% of returning user updates are malicious. TAG is a good choice for defense method regardless of the data distribution of its users.

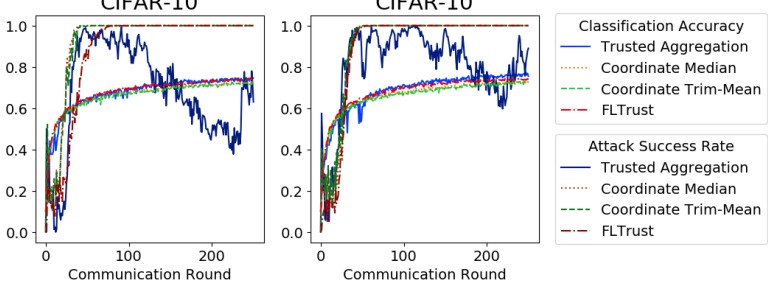

Figure 16: Model performance under DBA without (*left*) and with (*right*) Neurotoxin backdoor attacks with 40% malicious users and imbalanced local user data sets without tuning $\theta$.

## B.5 ROBUSTNESS TO TRUSTED USER DATA (CONTINUED)

### B.5.1 ROBUSTNESS TO DISTRIBUTION OF TRUSTED USER DATA

Recall that the trusted user either consists of a guaranteed non-malicious user or the centralized server collects a small validation data sets and acts as another user. When the trusted user is the centralized server, they should attempt to collect a balanced a representative data set. However, if the trusted user is an actual user or the centralized server fails to obtain such a representative sample, it is vital to understand how the data distribution of the trusted user impacts defense.

To better understand the effect of validation data distribution, we first omit finding the smallest scaling coefficient $\theta$ such that the global model obtains desirable classification accuracy. We begin by repeating the imbalanced user data experiments with the same scaling $\theta$ as before. Without a properly tuned $\theta$, TAG still effectively defends against combinations of DBA and Neurotoxin attacks when 10% (Figure 17) and 20% (Figure 18) of the selected user subset are malicious each round.

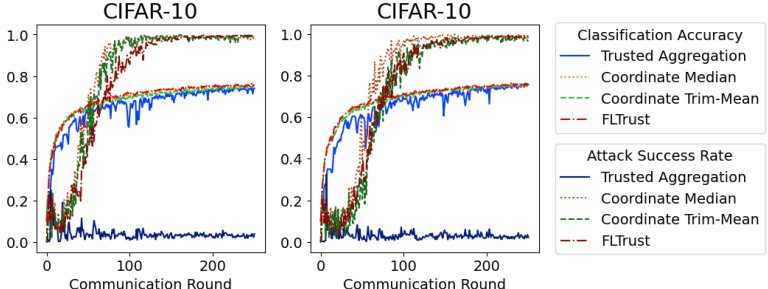

Figure 17: Model performance under standard (*left*) and Neurotoxin (*right*) backdoor attacks with 10% malicious users on imbalanced local user data sets with an equally imbalanced trusted validation data set without tuning $\theta$.

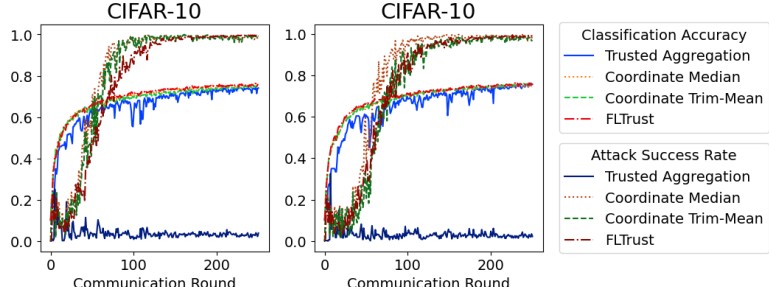

Figure 18: Model performance under DBA without (*left*) and with (*right*) Neurotoxin backdoor attacks with 20% malicious users on imbalanced local user data sets with an equally imbalanced trusted validation data set without tuning $\theta$.

TAG can even prevent the DBA and Neurotoxin attack with a 40% of attackers but now fails to prevent pure DBA, see Figure 19. We conclude that tuning the scaling coefficient $\theta$ is essential for every TAG defense application. Despite the success of TAG in our earlier attack settings, defending against backdoor attacks is a different task for differently distributed trusted user data. Careful tuning of $\theta$ will help to ensure the best protection TAG can provide against strong backdoor attacks.

See, Section 4.5 for proper experimentation with imbalanced users and trusted user using a tuned values for TAG's scaling coefficient $\theta$. With hyper-parameter tuning, TAG can be an effective defense against backdoor attacks regardless of whether or not any user has imbalanced data.

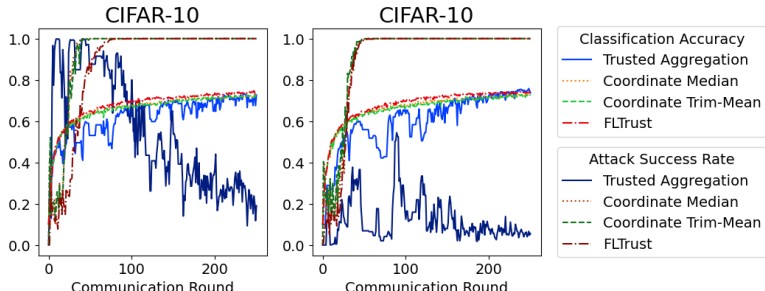

Figure 19: Model performance under DBA without (*left*) and with (*right*) Neurotoxin backdoor attacks with 40% malicious users on imbalanced local user data sets with an equally imbalanced trusted validation data set without tuning $\theta$.

### B.5.2 Robustness To Size Of Trusted User Data

Additionally, we want to determine whether our method depends on such size requirements. Hence we revisit the most substantial attack setting for the CIFAR10 data set but only allow the validation user to have a data set that is 20% of the size of the other local users. Note that for this experiment, all users have balanced and representative data. This experiment is most applicable to the case where the centralized server needs to collect data, especially for problems where the cost of obtaining data is high. Here the validation set is now only allowed 100 images, yet TAG prevents the backdoor attack and can achieve improved accuracy compared to the baseline robust aggregation methods. Hence, we conclude we do not need validation data of the same quantity as other local users to discriminate between benign and malicious returning models.

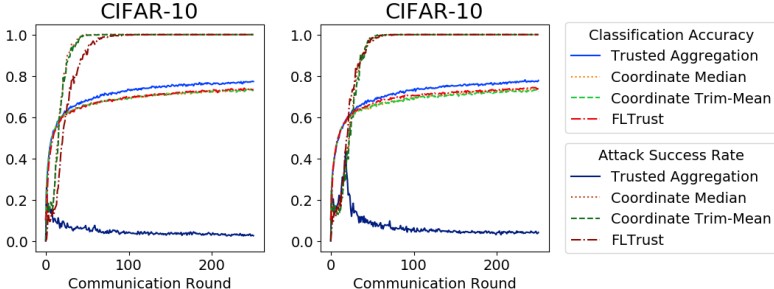

Figure 20: Model performance under DBA and Neurotoxin backdoor attacks with 40% malicious users on CIFAR10 where the validation data set is 20% the size of the local users. Still, the proposed method TAG performs well, and the other two aggregation methods do not work well in preventing any backdoor attack.

## C  SUPPLEMENTARY PROOFS

### C.1  PROOF OF PROPOSITION 1

$$\text{Assume } \forall c \in [1, ..., m], v^{(c)} \sim \text{Unif}(0, b_c)$$

$$\text{Define } V = \max_c \left( v^{(c)} \right) \text{ and } j = \arg\max_c (b_c)$$

$$v^{(j)} \leq V \leq b_j \implies E\left[ v^{(j)} \right] = \frac{b_j}{2} \leq E\left[ V \right] \leq b_j \implies E\left[ V \right] \leq b_j \leq E\left[ 2V \right]$$

### C.2  PROPOSITION 2

If we additionally assume independence between the class conditional distances, we can establish an additional lower bound, Proposition 2, for our largest possible benign change. Note, Proposition 2 converges asymptotically to the earlier lower bound as the size of the classification problem ($m \to \infty$) grows. Hence, this additional bound is provided to assist with choosing an appropriate $\theta$ for smaller-way classification problems.

**Proposition 2** *If $v^{(c)} \overset{\perp\!\!\!\perp}{\sim} Uniform(0, b_c), \forall c \in [1, \ldots, m]$ then $E\left[ \left( \frac{m+1}{m} \right) V \right] \leq b_j$ where $V = \max_c v^{(c)}$ and $j = \arg\max_c(b_c)$.*

$$\text{Assume } \forall c \in [1, ..., m], v^{(c)} \overset{\text{ind}}{\sim} \text{Unif}(0, b_j)$$

$$\text{Let } W = \max_c \left( w^{(c)} \right) \text{ where } w^{(c)} \overset{\text{iid}}{\sim} \text{Unif}(0, b_j) \implies \frac{W}{b_j} \sim \text{Beta}(m, 1) \implies E(W) = \frac{m}{m+1}$$

$$F_V(t) = P(V \leq t) = \prod_{i=1}^{m} P_{v^{(c)}}(t) \geq \prod_{i=1}^{m} P_{w^{(c)}}(t) = P(W \leq t) = F_W(t)$$

$$\implies E(V) = \int_0^\infty \left( 1 - F_V(t) \right) dt \leq \int_0^\infty \left( 1 - F_W(t) \right) dt = E(W)$$

$$\implies E\left[ \frac{V}{b_j} \right] \leq E\left[ \frac{W}{b_j} \right] = \frac{m}{m+1} \implies E\left[ \left( \frac{m+1}{m} \right) V \right] \leq b_j$$

