# OpenReview forum: "Trusted Aggregation (TAG): Model Filtering Backdoor Defense In Federated Learning"
_ICLR.cc/2023/Conference — Submitted to ICLR 2023_

### Official Review · Reviewer_qsGk · 2022-10-24

**Confidence:** 3
**Correctness:** 2
**Technical Novelty And Significance:** 3
**Empirical Novelty And Significance:** 3
**Recommendation:** 5

**Clarity, Quality, Novelty And Reproducibility:**

The paper is easy to follow, but the paper does not provide sufficient evidence to demonstrate the effectiveness of the proposed defense.

**Strength And Weaknesses:**

Strengths:
1. The paper finds an interesting observation that the backdoor attack generates different output distributions compared with clean models.
2. The proposed defense is effective against multiple backdoor attacks and can eliminate backdoor attacks even when 40% of clients are malicious.
3. The paper investigates the defensive effectiveness with different sizes of the validation dataset.

Weaknesses:
1. The proposed defense measures the distributional difference between clients to identify malicious updates. This defense assumes that all clients share similar data distributions.  However, data heterogeneity is a natural property in federated learning, where data distributions vary among clients. The paper only considers the imbalance data distribution with \alpha=1, which is not a strong non-iid case. It would be better to consider stronger non-iid data distributions.
2. The paper only considers some basic backdoor attacks and defenses. State-of-the-art attacks (e.g., edge case backdoors [1]) and defenses [2, 3, 4], should be evaluated/compared in the paper.
3. The proposed defense requires validation data on the server side to update the global model and use the global model to measure the distributional distances. This requires validation data to be similar to the clients’ data. It is nice to see the paper investigate the impact of the validation data size in the defense. It would be great if the paper could also investigate the different data distributions of the validation data.

[1] Wang, Hongyi, et al. "Attack of the tails: Yes, you really can backdoor federated learning." Advances in Neural Information Processing Systems 33 (2020): 16070-16084.

[2] Andreina, Sebastien, et al. "Baffle: Backdoor detection via feedback-based federated learning." 2021 IEEE 41st International Conference on Distributed Computing Systems (ICDCS). IEEE, 2021.

[3] Hayase, Jonathan, et al. "SPECTRE: defending against backdoor attacks using robust statistics." ICML, 2021.

[4] Tang, Di, et al. "Demon in the Variant: Statistical Analysis of DNNs for Robust Backdoor Contamination Detection." 30th USENIX Security Symposium (USENIX Security 21). 2021.


**Summary Of The Paper:**

The paper presents a backdoor defense for federated learning. The proposed defense is motivated by the differences in the output layer with and without backdoor attacks and aims to identify such changes to prevent backdoor attacks.

**Summary Of The Review:**

The paper investigates a timely problem of backdoor defenses in federated learning. The proposed defense is well-motivated. However, the comparison with the state-of-the-art attacks and defenses is missing in the paper. The paper does not consider the strong non-iid settings in federated learning.

---

> ### Author Response · Authors · 2022-11-18
> **Incorporating stronger defense comparison and validation data distribution studies**
>
> Dear Reviewer qsGk,
>
> First, although we added acknowledgment to a limitations section of more extreme non-iid settings, we want to make the case that with alpha = 1, we are considering a meaningful non-iid environment for federated learning. When users have significantly different data sets, producing a single shared model for such heterogenous tasks no longer makes sense.
>
> Second, we love your suggestion to investigate the data distribution of the trusted user! We have additional experimentation to show that when all users, including the trusted user, have data that is non-iid, TAG is still an effective defense even under our 10%, 20%, and 40% malicious update setting. We now have experimental results to support that TAG is robust to non-iid data distributions of all users.
>
> Next, we appreciate your suggestions for alternative defense baselines. However, per another reviewer’s request, we believe that FLTrust [1] is the best additional comparison we can include due to its specificity to federated learning and requires access to clean data. Also, FLTrust deserves your consideration as a state-of-the-ark defense. Its authors demonstrate its ability to defend against even adaptive attacks where the attacker knows the defense method used. However, for our attack settings, FLTrust, with adaptive success, fails to prevent even our mildest attack prevalence. Hence, we view our method’s vast outperformance of a similar approach with adaptive success as solid evidence of the quality of our proposed method.
>
> Regarding your attack suggestion, we do not believe Attack of the Tails [2] is relevant to our attack and evaluation framework. Instead, we are interested in proposing our method to defend against targeted backdoor attacks that can act on any input image instead of just edge cases.
>
> For defense suggestions, Spectre [3] and Demon [4] violate user data privacy, which is of utmost importance to federated learning. First, Spectre’s Algorithm would require filtering user data. Secondly, Demon [4] needs representations for all input images from a clean set and the training set that contains the attack. Next, Baffle [5] claims that “backdoors injected in early rounds are not durable.” However, DBA and Neurotoxin, which they do not compare to, have shown drastic improvements to attack longevity.
>
> Furthermore, “Baffle cannot be used from the beginning but, e.g., only after several hundreds of rounds, as otherwise many false positives will occur” [6]. On the other hand, TAG can be used starting in the first round and is effective against much stronger attack settings than considered by Baffle. As a result, we plan on comparing with Baffle as lower priority future work.
>
> We value your feedback and thank you for the great ideas. Also, we appreciate your constructive criticisms and their reconsideration, given our response.
>
> [1] FLTrust: Byzantine-robust Federated Learning via Trust Bootstrapping. arXiv:2012.13995 [cs.CR]
>
> [2] Attack of the Tails: Yes, You Really Can Backdoor Federated Learning. arXiv:2007.05084 [cs.LG]
>
> [3] SPECTRE: Defending Against Backdoor Attacks Using Robust Statistics. arXiv:2104.11315 [cs.LG]
>
> [4] Demon in the Variant: Statistical Analysis of DNNs for Robust Backdoor Contamination Detection. arXiv:1908.00686 [cs.CR]
>
> [5] BaFFLe: Backdoor detection via Feedback-based Federated Learning. arXiv:2011.02167 [cs.CR]
>
> [6] DeepSight: Mitigating Backdoor Attacks in Federated Learning Through Deep Model Inspection. arXiv:2201.00763 [cs.CR]

---

### Official Review · Reviewer_SvhM · 2022-10-24

**Confidence:** 4
**Correctness:** 3
**Technical Novelty And Significance:** 3
**Empirical Novelty And Significance:** 3
**Recommendation:** 5

**Clarity, Quality, Novelty And Reproducibility:**

The paper is well organized and well written. The proposed approach is novel, although the literature review is not very complete (important attacks and defenses in backdoors for federated learning are missing), which would be necessary to position the paper better, especially in terms of the threat model and the assumptions for the defender.
On the reproducibility side, as mentioned before, there are certain details about the experimental settings that are missing and that are important for the sake of reproducibility and to assess better the quality of the results.


**Strength And Weaknesses:**

Strengths:
+ Defending against backdoors in federated learning is a challenging problem of interest for the research community. The experimental results in the paper show that the proposed method (TAG) can mitigate the effect of state-of-the-art backdoor attacks such as Neurotoxin and DBA (Distributed Backdoor Attack).
+ Overall, the paper is well written and the authors explain well the intuition about the method they proposed.

Weaknesses:
+ In the experiments, the authors just compared with Coordinate Median and Trimmed Mean. This comparison is not really fair: on one side, these techniques do not assume the existence of a trusted validation dataset. On the other hand, these techniques are not specifically designed against backdoor attacks, but more general data and/or model poisoning attacks. It would be necessary to compare with existing defenses that rely on a similar set of assumptions, e.g. Fang et al. “Local Model Poisoning Attacks to Byzantine-Robust Federated Learning”, which uses trusted validation datasets.
+ The proposed method just looks at the last layer of the models. It is true that, for attacks like Neurotoxin and DBA this can be enough to mitigate the attack. However, this opens the door for adaptive attacks where the adversaries are aware of the defensive method applied. Then, the attackers can try to manipulate the parameters in the previous layers to try to evade detection. In this sense, the paper lacks an analysis of the robustness against adaptive attacks and possible limitations.
+ Some of the experimental settings used are not clearly described in the paper. For instance, the number of participants in the federated learning tasks, learning rates, optimizer, batch size, etc. I think this is especially important in the case of the number of participants, which is a factor that can play a critical role in the performance of the defense.


**Summary Of The Paper:**

The authors introduce a novel robust aggregation method for filtering backdoors in federated learning by using a trusted validation dataset. For this, the authors look at the differences in the distribution of the output layer for the different trusted and untrusted clients, removing those updates that deviate from the distribution in the trusted clients. The authors evaluate the proposed method against two state-of-the-art backdoor attacks in federated learning: Neurotoxin and the Distributed Backdoor Attack, showing the effectiveness of the proposed method to mitigate these attacks.

**Summary Of The Review:**

Although the assumption of having a trusted validation set/clients can be restrictive to certain applications, the algorithms proposed in the paper is interesting and seems to provide good performance against certain backdoor attacks. However, there are certain aspects that require a more thorough analysis:
+ The comparison with coordinate median and trimmed mean is not really fair as they do not assume having access to a trusted validation set or trusted clients. In this sense it is necessary to compare with existing defenses that make use of this assumption to really validate the benefits of the proposed approach and its possible limitations.
+ Looking just at the last layer can make the algorithm vulnerable to other attacks. Given this, I think that the evaluation against an adaptive attackers that is aware of this is necessary to have a more realistic evaluation of the proposed defense.
+ The description of the experimental settings used should be clearer and more detail.
+ Finally, the assumptions made in Proposition 1 and 2 about the uniform distribution of the class-conditional distances is not well justified and looks cherry picked. I think that the authors should justify better their assumptions in this sense.

---

> ### Author Response · Authors · 2022-11-18
> **Improving defense comparison and clarity**
>
> Dear Reviewer SvhM,
>
> We greatly appreciate your comprehensive feedback. As a result, we are now including a better baseline comparison and addressing adaptive attacks. For our additional baseline, we have FLTrust [1], per another reviewer’s request, in all our experiments. FLTrust builds upon the work [2] you reference, requires clean data and has demonstrated success against adaptive attacks.
>
> We see that for our attack settings, FLTrust, with adaptive success, fails to prevent even our mildest attack prevalence. Hence, we view our method’s vast outperformance of a similar approach with adaptive success as solid evidence of the quality of our proposed method. However, we include a limitations section that explicitly states how a malicious user should perform an adaptive attack against our defense and acknowledges that additional work is needed to extend the capabilities of our defense. Meanwhile, adaptive attack experiments are in progress.
>
> Furthermore, we have made several changes to improve the transparency and accuracy we communicate our findings. First, we now explicitly include a table in the appendix identifying all hyperparameters for our results. In combination with a GitHub repository, we are confident that others can replicate and extend our results. Also, we needed to improve our language regarding our Proposition to communicate our intentions correctly. The sole purpose of our Propositions was to assist in restricting the search space for an optimal scaling coefficient theta. However, we agree that the Propositions’ assumptions should not hold in general, being highly dependent on all hyperparameters and model architecture. Therefore, to better communicate our original intentions, we modify our Algorithm to theta >= 1 and make significant changes to the language used when presenting these facts.
>
> As a result of your comments, our work communicates our method with better accuracy and transparency. Thank you for the time and care you spent constructing your initial review and reconsideration based on our response.
>
> [1] FLTrust: Byzantine-robust Federated Learning via Trust Bootstrapping. arXiv:2012.13995 [cs.CR]
>
> [2] Local Model Poisoning Attacks to Byzantine-Robust Federated Learning. arXiv:1911.11815 [cs.CR]

---

> > ### Comment · Reviewer_SvhM · 2022-11-22
> > **Comments after rebuttal**
> >
> > Thank you very much for your responses and clarifications.
> > After reading all the comments from the authors and the other reviewers, I think that the method proposed in the paper looks interesting but, as pointed out by the reviewers, there are still some points that require a bit more work. Thus, I'm keeping my score.
> > Nevertheless, I think that the paper has potential and I'd like to encourage the authors to address the concerns pointed out by the reviewers.

---

### Official Review · Reviewer_JmoH · 2022-10-25

**Confidence:** 4
**Correctness:** 2
**Technical Novelty And Significance:** 2
**Empirical Novelty And Significance:** 2
**Recommendation:** 3

**Clarity, Quality, Novelty And Reproducibility:**

An adaptive attack is missing. The proposed defense is based on the observation
in Section 3. Is the proposed detection method still effective under adaptive
attacks (the attacker is aware of the defense and tries to bypass it)? During
poisoning, a possible adaptive attack constrains the backdoored models' final
layer output distributions. In other words, making the output distributions of
the backdoored models similar to that of benign models. Goldwasser et al.
(Planting Undetectable Backdoors in Machine Learning Models) show that it is
possible to do so. In practice, Doan et al. NeurIPS 2021 have also shown that backdoors can be imperceptible from both the input and latent spaces.

In this paper, the proposed defense method is based on the observation in
Figure 1. Unfortunately, the details of the key experiments are missing. For
example, the used datasets, models, and attacks are all unclear. The empirical
results are also limited to a single setting. Thus, it is unclear to me whether the conclusion is reasonable.

The definition of the "output layer" is unclear. It lacks a discussion about
how to select the output layer. In Figure 1, this paper uses the final hidden
layer as the output layer. It is unclear why the final hidden layer is
selected. The results of using other layers are missing.

Comparisons with some related works are missing. Only two baseline defenses
(i.e., Median and Trim-mean) are involved in the experiments. It lacks
comparisons with related works [1,2]. Therefore, it is hard to say if the
proposed method achieves better performance than existing methods.
Generalizations to different models are unclear. Existing defense [1]
demonstrated it can generalize to different models. However, all experiments
in this paper are conducted on a single model, i.e., ResNet18. Thus, the
generalization of the proposed method to different model architectures (e.g.,
VGG, DenseNet, and ViT) are unclear.

[1] Rieger et al., Deepsight: Mitigating backdoor attacks in federated
learning through deep model inspection. NDSS 2022.
[2] Nguyen et al., FLAME: Taming Backdoors in Federated Learning. USENIX
Security 2022.

**Strength And Weaknesses:**

Strengths

· Interesting topic.

Weaknesses

· Comparisons with some related works are missing.
· Generalization to different models is unclear.
· Discussion about the potential adaptive attacks is missing.
· Details about the experiments in Figure 1 are unclear.
· How to select the output layer is unclear.
· The writing needs improvement. Details of the critical experiments (e.g., Figure
1) is unclear.

**Summary Of The Paper:**

This paper proposes a defense against backdoor attacks on federated learning.
The method works by detecting malicious users by analyzing the final layers'
output. The proposed method is based on observing that the distributions of
the final layer's output between the malicious and benign models differ.
Experiments on CIFAR-10, CIFAR-100, and STL-10 demonstrate that the proposed
method is effective.

**Summary Of The Review:**

Details of the critical experiments are unclear. The evaluation is also weak since
it only uses a single model and lacks a discussion about adaptive
attacks. Discussion on existing attacks is also missing.

---

> ### Author Response · Authors · 2022-11-18
> **Addressing reproducibility, model generalizability, and additional comparisons**
>
> Dear Reviewer JmoH,
>
> First, our experiments’ transparency and reproducibility are of utmost importance. Therefore, per your suggestion, we now explicitly include a table in the appendix identifying all hyperparameters for our results. In combination with a GitHub repository, we are confident that others can replicate and extend our results. Furthermore, to address model generalizability, we repeat our main results using VGG to demonstrate that our conclusions do not depend on our model choice.
>
> Second, regarding our method, you helped us identify language that we needed to improve significantly in our work (including the incorrect caption for Figure 1). As a result, we now clearly communicate that our defense operates on predicted scores, the model’s output, before any softmax operation for classification. Again, we apologize for any confusion.
>
> Deepsight [1] and FLAME [2] are recent works without publicly available code. We plan on comparing them with our method in the future. In the meantime, we will incorporate them into our related work section and instead include FLTrust [3] as an additional baseline defense, per another reviewer’s suggestion.
>
> We believe that a comparison with FLTrust is more relevant to our work. First, FLTrust also requires clean data. Secondly, FLTrust demonstrated success against adaptive attacks. However, FLTrust, which has adaptive success, fails to prevent even the mildest attack setting we consider in our paper. Thus, our method’s vast outperformance of a similar approach with adaptive success is strong evidence of the quality of our proposed method. We also include a limitations section that explicitly states how adaptive attacks should attack our defense and acknowledge such attacks as a limitation. Meanwhile, adaptive attack experiments are in progress.
>
> We hope our clarificatons help with any misunderstanding we created in communicating our method. Again, we thank you for your advice and careful reconsideration of your score, considering recent changes.
>
> [1] DeepSight: Mitigating Backdoor Attacks in Federated Learning Through Deep Model Inspection. arXiv:2201.00763 [cs.CR]
>
> [2] FLAME: Taming Backdoors in Federated Learning. arXiv:2101.02281 [cs.CR]
>
> [3] FLTrust: Byzantine-robust Federated Learning via Trust Bootstrapping. In NDSS, 2021.

---

### Official Review · Reviewer_zmnT · 2022-10-27

**Confidence:** 4
**Correctness:** 3
**Technical Novelty And Significance:** 1
**Empirical Novelty And Significance:** 1
**Recommendation:** 3

**Clarity, Quality, Novelty And Reproducibility:**

The paper has limited quality and novelty.

1. When a clean dataset is available, many methods can be used to defend against backdoor attacks, but the paper does not compare with any of them. e.g., Neural Cleanse [A] can be used to detect backdoored local model. Also, FLTrust [B] can be used to defend against backdoor attacks. The authors are suggested to compare with these methods.

2. Adaptive backdoor attacks are not considered. What happens if an attacker knows your defense? How can an attacker adapt its backdoor attack to your defense? It is important to evaluate adaptive attacks since the proposed method does not have certified security guarantees.


3. How does the clean dataset distribution impact the performance of the proposed method?

[A] Neural Cleanse: Identifying and Mitigating Backdoor Attacks in Neural Networks. In IEEE S&P, 2019.

[B] FLTrust: Byzantine-robust Federated Learning via Trust Bootstrapping. In NDSS, 2021.


**Strength And Weaknesses:**

+ Backdoor attack is a severe threat to FL, and thus defending against such attack is relevant.

**Summary Of The Paper:**

This paper proposes a defense against backdoor attacks to FL. The assumption is that the system has access to some clean validation dataset, which can be used to filter backdoored model updates from malicious clients via analyzing the output layer distribution. Some evaluation is performed to show the effectiveness of the defense against existing backdoor attacks, and comparison is performed for simple baselines.

**Summary Of The Review:**

The paper studies an important topic, but the novelty and quality are limited.

---

> ### Author Response · Authors · 2022-11-18
> **Our improvements to clarity, quality, and reproducibility**
>
> Dear Reviewer zmnT,
>
> You gave excellent suggestions for an additional backdoor defense to compare with our method.
> FLTrust [1] is an essential extra baseline due to its similarity (requiring clean data) with our method. Hence, we now include and vastly outperform FLTrust in every single experimental result of our paper. Although FLTrust demonstrated great success against adaptive data poisoning attacks in the original article, it fails to prevent even the mildest attack setting we consider in our paper. Nevertheless, our method’s vast outperformance of a similar approach with adaptive success is strong evidence of the quality of our proposed method. However, we have now included an explicit limitations section where we acknowledge and suggest precisely how a malicious user should attack our defense in the adaptive attack setting. Meanwhile, we have adaptive attack experimentation in progress.
>
> However, we do not think Neural Cleanse [2] is relevant for federated learning. Their Algorithm for detecting backdoor attacks relies upon obtaining a reverse-engineered trigger. This procedure imposes structural assumptions on the backdoor manipulation and requires solving a minimization problem for every user, which is not scalable to real-world problems. On the other hand, our method only requires a single forward pass for each user on a small, trusted data set. In summary, our proposed method does not require any assumptions on the backdoor attack and is computationally trivial in contrast to the application of Neural Cleanse to federated learning.
>
> Another excellent suggestion is investigating the impact of the data distribution on the trusted user. We now include experimentation to show that our method is effective at various sizes and distributions for the data set of the trusted user. Our proposed defense is still effective when 10%, 20%, or 40% of updates are malicious, and the trusted data set is non-iid. Furthermore, FLTrust is ineffective against the considered attacks even when the dataset is balanced and of full size.
>
> Your comments inspired several substantial revisions to our article, which we believe have greatly improved the quality of this project. So again, we greatly thank you for your assistance and appreciate your careful reconsideration given our modifications.
>
> [1] FLTrust: Byzantine-robust Federated Learning via Trust Bootstrapping. In NDSS, 2021
>
> [2] Neural Cleanse: Identifying and Mitigating Backdoor Attacks in Neural Networks. DOI: 10.1109/SP.2019.00031

---

### Decision · Program_Chairs · 2023-01-20

**Decision:**

Reject

**Justification For Why Not Higher Score:**

Major concerns remained form all reviewers, e.g. on fairness of comparisons

**Justification For Why Not Lower Score:**

N/A

**Metareview: Summary, Strengths And Weaknesses:**

The paper studies defenses against backdoor attacks to federated learning, which is an important topic.

Unfortunately many concerns remained from the reviews both on the level of novelty and quality of the results. This includes key assumptions made, and also the selection of baseline defenses to compare to, and allowing them access to a trusted validation set.

We hope the detailed feedback helps to strengthen the paper for a future occasion.